# Beekeepers' perceptions toward a new omics tool for monitoring bee health in Europe

Elena Cini[1,2]*, Simon G. Potts[1], Deepa Senapathi[1], Matthias Albrecht[3], Karim Arafah[4], Dalel Askri[4], Michel Bocquet[5], Philippe Bulet[6], Cecilia Costa[7], Pilar De la Rúa[8], Alexandra-Maria Klein[9], Anina Knauer[3], Marika Mänd[10], Risto Raimets[10], Oliver Schweiger[11,12], Jane C. Stout[13], Tom D. Breeze[1]*

1 Centre for Agri-Environmental Research, School of Agriculture, Policy and Development, University of Reading, Reading, England, United Kingdom, 2 School of Environmental and Natural Sciences, Bangor University, Bangor, Wales, United Kingdom, 3 Agroecology and Environment, Agroscope, Zurich, Switzerland, 4 Plateforme BioPark d'Archamps, Archamps, France, 5 Apimedia, Pringy, Annecy, France, 6 Institute for Advanced Biosciences, CR Inserm U1209, CNRS UMR5309, Université Grenoble Alpes, Team-Verdel: ARN, Epigénétique et Stress/RNA, Epigenetics and Stress, Grenoble, France, 7 CREA Research Centre for Agriculture and Environment, Bologna, Italy, 8 Department of Zoology and Physical Anthropology, Faculty of Veterinary, University of Murcia, Murcia, Spain, 9 Chair of Nature Conservation and Landscape Ecology, University of Freiburg, Freiburg, Germany, 10 Institute of Agricultural and Environmental Sciences, Estonian University of Life Sciences, Tartu, Estonia, 11 UFZ–Helmholtz Centre for Environmental Research, Department of Community Ecology, Halle, Germany, 12 German Centre for Integrative Biodiversity Research (iDiv) Halle-Jena-Leipzig, Deutscher, Leipzig, Germany, 13 Trinity College Dublin, School of Natural Sciences, Botany Department, College Green, Dublin, Ireland

* elena.cini.ec@gmail.com (EC); t.d.breeze@reading.ac.uk (TDB)

**Data Availability Statement:** The full, anonymised dataset is available from Zenodo (DOI: 10.5281/zenodo.10245699).

## Abstract

Pressures on honey bee health have substantially increased both colony mortality and beekeepers' costs for hive management across Europe. Although technological advances could offer cost-effective solutions to these challenges, there is little research into the incentives and barriers to technological adoption by beekeepers in Europe. Our study is the first to investigate beekeepers' willingness to adopt the Bee Health Card, a molecular diagnostic tool developed within the PoshBee EU project which can rapidly assess bee health by monitoring molecular changes in bees. The Bee Health Card, based on MALDI BeeTyping®, is currently on level six of the Technology Readiness Level scale, meaning that the technology has been demonstrated in relevant environments. Using an on-line survey from seven European countries, we show that beekeepers recognise the potential for the tool to improve colony health, and that targeted economic incentives, such as subsidises, may help reduce cost being a barrier to the adoption and frequent use of the tool. Based on the description of the tool, 43% of beekeepers appear to be moderately confident in the effectiveness of the Bee Health Card. This confidence could increase if the tool was easy to use and not time consuming, and a higher confidence could also contribute to raising the probability of accepting extra costs linked to it. We estimate that, in the worst-case scenario, the cost per single use of the Bee Health Card should be between €47–90 across a range of European countries, depending on the labour and postage costs. However, the monetary benefits in terms of honey production could exceed this. In order to successfully tackle colony health issues, it is recommended using the BHC five times per year, from the end to the beginning

**Funding:** The PoshBee project (https://poshbee.eu/) received funding from the Horizon Europe EU research and innovation programme (https://research-and-innovation.ec.europa.eu/) under grant agreement No 773921 (https://cordis.europa.eu/project/id/773921). The funders had no role in study design, data collection and analysis, decision to publish, or preparation of the manuscript.

**Competing interests:** The authors have declared that no competing interests exist.

of winter. Finally, we discuss the knowledge needs for assessing beekeeper health tools in future research.

## Introduction

Pollination is a key ecosystem service benefitting about 75% leading food crops worldwide [1]. Animal pollination in particular is estimated to provide global food crops with benefits worth between $235–577 bn per year [2]. The most common crop pollinators globally are solitary and social bees [3, 4], and although wild bees can be more efficient and better contribute to crop pollination [5], managed honey bees (*Apis* spp.) are crucial in many commercial crop systems. In fact, they are suitable for a wide range of mass-flowering crops grown in intensive systems due to their large colony sizes, recruitment behaviour, and strongly generalistic foraging behaviour. Globally, they are estimated to visit more than 50% of animal-pollinated crops [6].

Both wild and managed pollinators are threatened by a range of pressures, and given their important role in the ecosystem, there is global concern over the potential impacts of their continued decline [7]. In particular, agricultural intensification is leading to the loss of diverse flowering resources in favour of low-diversity habitats such as intensive grasslands [8] and mass-flowering crops, which are increasingly exposing bees to agrochemicals [9, 10] with a range of lethal and sub-lethal effects on their health [11], and depriving bees of suitable nesting and foraging resources [9]. Moreover, the commercialisation of managed bees and shifts in climate are also increasing the spread of disease and pathogens such as *Varroa destructor* and associated viruses [12] and their spill-over effects on wild pollinators [13–15], as well as the invasion of alien species such as the Asian hornet (*Vespa velutina)*, unintentionally introduced to Europe from Asia [16, 17].

As a result of these pressures, several European studies have reported high incidences of honey bee colony losses [18, 19] and growing rates of colony health disorders [20, 21]. The higher incidence of health issues has led to greatly increased costs for beekeepers [19, 22]. Since such costs are only seldom recuperated through pollination activities [22], available honey bee stocks may be incapable of matching the pace of expansion on animal-pollinated crops [23].

To address such issues on managed honey bee colonies, the European Union has provided direct support through various national honey bee health programmes (*e.g.*, Apiculture programmes [24]) and surveillance measures (*e.g.*, EU Reference Laboratory [25]), and indirect support through agri-environment schemes that can promote pollinator-friendly practices [13, 26]. The EU is also supporting research programmes involving new technologies to monitor the health of beehives, such as the 'Swarmonitor' [27] and the PoshBee project [28]. However, research into opinions, attitudes, and/or management practices of European beekeepers is limited to few studies [22, 29–32] and to date, no study has looked into their willingness to adopt these new tools. By contrast, numerous studies on farmers and their attitudes toward the use of novel technologies show that, despite the purported benefits, perceived costs and complexity use can discourage their uptake [33–35]. In light of the rise of bee health issues in Europe, it is therefore important to understand what barriers and incentives beekeepers perceive when it comes to adopting novel technologies.

Here we present the results of a questionnaire survey carried out in seven European countries, exploring beekeepers' perceptions of the Bee Health Card (BHC), a laboratory tool developed and tested as part of PoshBee (Horizon EU [28]). This tool was designed to assess a range

of biotic and abiotic stressors by monitoring molecular changes in bee and beehive products through the MALDI BeeTyping® Mass Spectrometry technology [36, 37]. It works by detecting molecular changes linked to altered health status to identify the impact of pesticides on bees, including insecticides, herbicides, and fungicides. Moreover, the possibility of detecting peptides or proteins that may be released by viruses, pathogens, or parasites (including *Varroa destructor*, *Nosema* spp., and *Crithidia* spp.) is currently being explored. The ultimate aim of the BHC is to reduce the risk of colony mortality by providing early warnings to beekeepers on risks to colony health, enabling the adoption of timely mitigation measures and contributing to the productivity of beehives and the provision of crop pollination services.

As the rate of adoption may influence the wider benefits of technologies [38], understanding the factors that may incentivise or hinder their uptake is crucially important. Hence, we developed an analytical framework for estimating the potential costs and benefits of the Bee Health Card, spanning the following three steps: (i) assessing the willingness to use the BHC and the rate at which it could be used, (ii) completing a comprehensive analysis of the costs of using the BHC, and (iii) comparing these with the potential economic benefits to beekeepers (avoiding winter losses) and to the whole apicultural sector (honey production maintained).

## Methodology

### Beekeepers survey

An online questionnaire survey was created using Qualtrics [39] to explore what benefits and barriers could encourage or discourage beekeepers to use the Bee Health Card (BHC), and how to better support its wide uptake.

The survey was anonymous, and it was approved by the University of Reading Ethics Committee. When opening the survey link, respondents were informed through a participant information sheet that, if they decided to fill out the survey, they would consent to the terms of data storage and use approved by the committee.

The survey was translated (S1 Appendix) and distributed to beekeepers of seven European countries involved in a large-scale fieldwork conducted within the PoshBee project [40]: Estonia, Germany, Ireland, Italy, Spain, Switzerland, and the UK. Germany, Italy, and Spain are among the EU countries with the highest honey production [41], as well as the largest agricultural production [42].

Prior to distribution, the survey was peer reviewed by researchers and/or beekeepers from these countries. It was then promoted through the PoshBee social media channels, its website, and various beekeeping associations and magazines (summarised in S1 Appendix). Such participant recruitments took place from 30th July 2020 to 31st January 2021.

The BHC [43] is currently a Technology Readiness Level 6 (*i.e.*, the technology has been demonstrated in relevant environments [44]), and its effectiveness at a large scale has not yet been directly evaluated. Therefore, an infographic was designed to introduce respondents to the tool and how it works, based on insights given by BIOP (BioPark Archamps) and CNRS (Centre National de la Recherche Scientifique), who are leading its development (Fig 1).

The structure of the survey is reported in Table 1, whilst the full survey can be found in S1 Appendix.

### Costs associated with the Bee Health Card

The cost of the tool was estimated using information on the material and labour requirements from the tool development trials [45, 46]. Staff cost was drawn from the EU Pollinator Monitoring Scheme analysis [47], while material cost was taken from Fisher Scientific UK and converted to Euros using the 2020 annual average exchange rate from the European Central Bank

## Bee Health Card

The Bee Health Card is a tool under development which will allow beekeepers and veterinarians to have a rapid insight into the health of their colonies. Beekeepers will send a sample of live bees to a laboratory which will assess the exposure of the bees to pesticides, diseases, parasites, and malnutrition. The laboratory will then send the beekeeper an electronic report with information on the health status of their bees, and what is likely to be affecting the colony; it will help inform beekeepers and veterinarians when choosing appropriate medicinal treatments for their colonies.

The expected time window between the shipment and the results is 4-6 days. A business plan to define the tool cost is currently under development, but it should be below 25€ (22£).

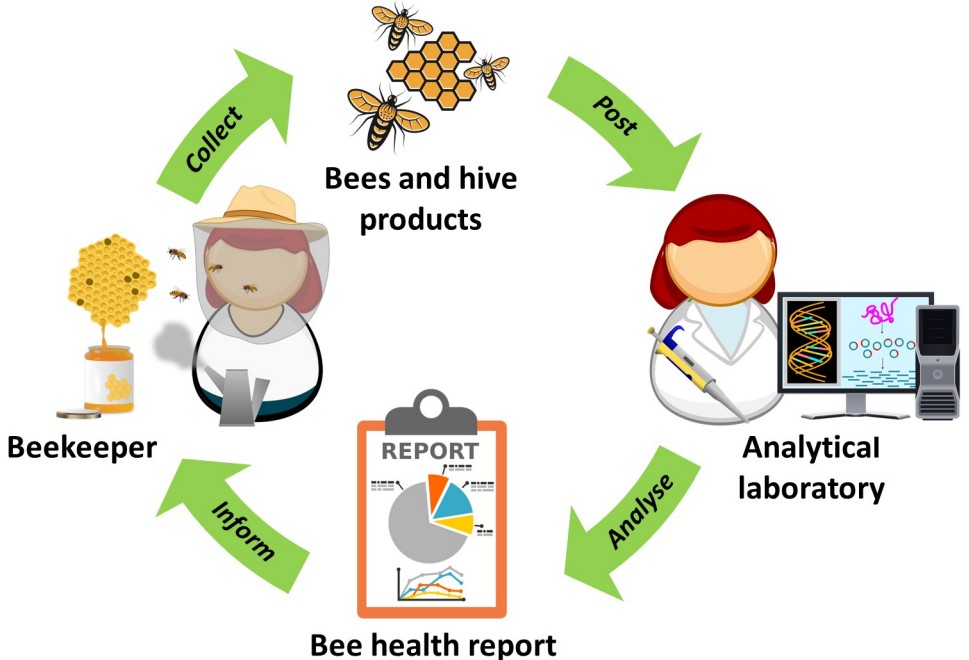

**Fig 1. Bee Health Card infographic shown to surveyed respondents.**

(ECB) [48] (S2 Appendix). Assuming each country would have a competent laboratory to analyse samples, postage cost was calculated as an average between TNT and DHL local couriers using a 1kg parcel at 32 x 24 x 15 cm as a guideline package with next-day delivery (see TNT.com and DHL.com). ECB exchange rates were used to convert British Pounds and Swiss Francs into Euros as appropriate [48].

A full-time administrator was assumed for each country, and overheads were assumed to be 1.5 times the cost of staff members, reflecting a typical high-rate overhead from a university or other research organisation. Only the variable costs involved in the BHC were estimated, as fixed laboratory costs will depend on the structure of the BHC scheme and the scale of the

Table 1. **Summary of survey questions.** See Table A in S1 Appendix for full survey.

| Topic | Questions overview |
|---|---|
| Beekeeping activity | Beekeeping experience, reasons to practice it, and communication with growers |
| Bee health | Important sources of information on beehive health, beehive health checks |
| Bee decline | Opinions and interest about bee decline and how to reduce it |
| Bee Health Card | Potential barriers to and benefits of the BHC, willingness to adopt it, and frequency of use in two scenarios: one with economic incentives (*e.g.*, subsidies, grants, certified products), and one without |

operation. However, a list of chemicals and items of laboratory equipment is provided in S2 Appendix, split into collection cost (*i.e.*, the costs of kit required to collect the samples) and laboratory cost (*i.e.*, the costs of analysing the data).

The number of samples to post was based on the total number of beekeepers in each country multiplied by the relative willingness to adopt the tool among respondents to the survey (see Table 6 in 'Results'). The estimated number of beekeepers for EU countries were taken from National Apicultural Programmes [49–53], as these are considered the most authoritative estimates, while for the UK and Switzerland (extra-EU) it was drawn from Gray et al. [21] (Table 2).

The number of laboratory staff required to run the BHC was estimated as one technician per 51,000 samples (200 samples per day x 255 working days/year). We assumed that each beekeeper would utilise the BHC only once per year to act as a minimum baseline for these costs (although the recommended yearly use is higher–see Discussion).

### Estimating Bee Health Card benefits

Since the effect of the BHC on colony survival has not been empirically tested yet, we explored the potential effectiveness using a hypothetical 50% overall increase in national colony survival if the BHC is used at an optimal rate. However, the actual effectiveness at a national scale may vary depending on the relationship between the adoption rate ($U$) and maximum possible effectiveness ($E$)–if the tool is capable of detecting health problems quickly enough to reduce pest and disease transmission and influence wider management before health issues become severe, it could be effective even at low levels of adoption. Alternatively, if the tool cannot identify issues as quickly, a higher use may be required for widespread effectiveness. Therefore, three different 'efficiency frontiers' were applied:

a. Linear: the tool is equally effective regardless of how many beekeepers use it, thus a 1% increase in use generates an E/100% increase in effectiveness. Here, 50% of the maximum

Table 2. **Estimated number of beekeepers per country.**

| Country | Beekeepers | Source |
|---|---:|---|
| Estonia | 5,215 | [49] |
| Germany | 11,600 | [50] |
| Ireland | 3,300 | [51] |
| Italy | 56,059 | [52] |
| Spain | 28,786 | [53] |
| Switzerland | 18,150 | [21] |
| UK | 39,475 | [21] |

efficiency is met at 50% adoption.

$$\sum_{f=1}^{F} E_f = U_f \times MaxE_f$$

Where:
E = effectiveness, f = frequency of use, U = adoption rate, MaxE = maximum regional effectiveness (50% overall increase in colony survival) at 100% adoption.

b.  Pessimistic: the tool is only effective at a large scale if it is widely adopted, meaning that most beekeepers would need to use it before it produces effective management. Here, 50% of the maximum efficiency is met at ~70% adoption.

$$\sum_{f=1}^{F} E_f = U_f^{2} \times MaxE_f$$

c.  Optimistic: the tool is very effective even at smaller adoption rates, hence even if only a few beekeepers use it, it can quickly detect widespread disease issues and control them before they result in widespread losses even among non-users. Here, 50% of the maximum efficiency is met at ~30% adoption.

$$\sum_{f=1}^{F} E_f = [U_f \times (2 - U_f)] \times MaxE_f$$

Each of these frontiers was applied to estimate the total number of additional colonies that survive winter (S) in the seven countries studied (c), using (i) the adoption frequency of the tool from each country (based on the stated rate of adoption–fc), derived from the percentage of beekeepers willing to use the tool in a best-case (economic incentives, no extra costs) and worst-case (no economic incentives, with extra costs) rate of adoption scenarios, (ii) national colony winter loss rates (L) from Gray et al. [21], and (iii) total national colony numbers (N), taken from FAOSTAT [54], NBU [55], and EC [51] (Table 3).

$$S_c = N_c \times L_c - [N_c \times (L_c \times E_{fc})]$$

Where:
S = Number of additional surviving colonies, c = country, N = Total colony numbers, L = winter loss rate (%), $E_{fc}$ = efficiency at the rate of adoption from respondents in each country.

**Table 3. Rate of colony losses and colony numbers.**

|  | Estonia | Germany | Ireland | Italy | Spain | Switzerland | UK |
|---|---|---|---|---|---|---|---|
| Total colonies[1] | 48,720 | 771,850 | 22,278[3] | 423,144 | 2,901,680 | 179,473 | 255,000[2] |
| Winter losses[3] | 8.3% | 11.6% | 3.9% | 8.8% | 17.6% | 7.4% | 5.4%[4] |

[1]Based on the average of 2016–2020 from FAOSTAT [54]. [2]As no FAO data were available for the UK in the selected time period, the average number of hives from 2017–2020 estimated by the National bee Unit [55] were used instead. [3]As colony numbers are not reported to the FAO, we used the average of the last two years reported in Annex 3 of the Ireland National Apicultural programme [51]. [4]Winter losses excluded the proportion due to queen failures and natural disasters. [4]Author calculation based on the data in Gray et al. [21] for each constituent country.

Finally, we estimated the economic benefits of colony survival. Ideally, this would be based on the direct costs of replacing beehives, but this has rarely been studied and will likely vary significantly at national and subnational levels [22]. Instead, economic benefit is quantified via the possible change in national scale honey output. Reducing national colony losses would consequently lead to an increase in the overall honey production per hive (H); since there is no definitive assessment of this effect (P), we used a conservative estimate of 20% difference in honey production between new and established colonies, based on the difference in median between 6 and 8 combs in Gąbka et al. [56].

$$E_c = S_c \times \left( \frac{H_c}{N_c} \right) \times P$$

Where:

E = market value of additional honey per hive in €, c = country, H = total value of honey production in € from FAOSTAT [54], N = total number of colonies, S = number of additional surviving colonies, P = percentage difference in honey production between new and established colonies.

## Statistical analysis

Correlations among answers were explored using Kendall Rank Correlation Analysis. Given the very high number of correlations, two Multiple Correspondence Analyses (MCA) [57] were conducted to identify groups of benefits and barriers that could be clustered to reduce the number of variables for later use in regression analyses. MCA is commonly and efficiently used with survey responses to visualise and cluster categorical data based on their proximity on the plot, allowing to explore the correspondence between numerous variables at once (*e.g.*, see [58–61]). The first MCA (MCA 1) was performed with variables related to the willingness to use the BHC and accept extra costs with and without economic incentives, and the second MCA (MCA 2) was conducted with variables related to the use frequency of the BHC with and without economic incentives. To reduce the number of categories shown on the MCA plots and avoid very polarised results (*e.g.*, very few people strongly agreeing with a statement, but many agreeing with it), we grouped (i) 'strongly agree' with 'agree' answers, (ii) 'strongly disagree' with 'disagree', (iii) 'extremely confident' with 'very confident', and (iv) 'moderately confident' with 'slightly confident'. We then proceeded to cluster different variables based on their proximity on the plot; variables of the same type (*i.e.*, benefits with benefits, barriers with barriers), and with 'agree', 'disagree', and 'neutral' answers grouping in the same way on the plot, were clustered together.

Afterwards, six binary logistic regression analyses were performed to investigate (i) the willingness to use the BHC with incentives, (ii) the willingness to use the BHC without incentives, (iii) the willingness to accept its extra costs with incentives, (iv) the willingness to accept its extra costs without incentives, (v) its frequency of use with incentives, and (vi) its frequency of use without incentives.

Due to low frequencies of some responses across all countries (*i.e.*, 'never' = four answers (0.7%) with incentives and eight answers (1.9%) without incentives; 'regular use' = 52 answers (12.6%) without incentives), we merged the use frequency (response variable) into two categories: (i) 'more frequent use', including respondents who would use the BHC somewhat frequently (either monthly or more irregularly, but always a few times per year), and (ii) 'limited to no use', comprising beekeepers who would use the tool just with a reasonable suspicion, or never. Respondents who indicated they were not interested in the BHC, either with or without incentives, were not shown the corresponding use frequency questions, resulting in 51 and 62 missing values for frequency with and without incentives, respectively. To ensure the analyses accurately reflected respondents'

preferences for using the BHC, and to avoid reducing the sample size by excluding responses, these missing values were treated as 'never' and categorised under 'limited to no use'.

For each of the six response variables listed above, the clusters obtained from the MCAs and the rest of the unclustered variables shown on the plots (see Table 4) were used as categorical predictors. To understand the tendency of each respondent to disagree, be neutral, or agree with the variables grouped in one cluster, an average score was calculated for each cluster attributing a score of 1 to 'disagree', 2 to 'neutral', and 3 to 'agree' to its variables. Decimals were then rounded to the nearest integer, and the answer was labelled as 'disagree' if the average score of the cluster was 1, 'neutral' if it was 2, and 'agree' if it was 3.

After creating the global models, the Variance Inflation Factor (VIF) of each term was checked to make sure only VIF $\leq 5.0$ were left in the model to avoid multicollinearity issues [62]. Model selection was based on a backward stepwise approach, removing terms with the highest p-value one by one until only significant terms ($p < 0.05$) were left in the models (S4 Appendix). Analyses were performed using RStudio (R version 4.2.3) [63].

## Results

### Sample description

We received 827 responses, of which 474 were usable (completion rate = 57.3%). The usable responses across the survey network varied substantially, with beekeepers from Ireland and the UK comprising more than 50% of all respondents (Table 5 below and S3 Appendix).

**Table 4. Clusters and individual variables shown on MCA plots.** The plots investigate (i) the willingness to use the tool and accept extra costs (MCA 1), and (ii) the frequency of use of the tool (MCA 2), with and without economic incentives. The way each variable and cluster was used in the subsequent regression analysis ('Regression') is also reported. See Table M in S3 Appendix for number and percentages of beekeepers agreeing, being neutral, and disagreeing with each cluster.

| Codes | Variables corresponding to codes | Classification | Plot | Regression |
|---|---|---|---|---|
| Cluster 1 (cp_ben + bh_ben + g_ben + qe_ben + ep_ben + pp_ben + p_ben + tc_ben) | Improved crop pollination + improved bee health + better communication with growers + tool quick and easy to use + environment protection + pollinator protection + higher productivity + lower treatment cost | Benefits | MCA 1, MCA 2 | Predictor |
| Cluster 2 (t_bar+ d_bar + i_bar) | Time-consuming + difficult + not important | Barriers | MCA 1, MCA 2 | Predictor |
| c_bar | Cost | Barriers | MCA 1, MCA 2 | Predictor |
| e_bar | Not effective | Barriers | | |
| g_bar | Poor communication with growers | Barriers | MCA 1, MCA 2 | Predictor |
| evc, msc, nc | Confidence in the effectiveness of the tool (extremely to very confident, moderately to slightly confident, not confident) | Effectiveness | MCA 1, MCA 2 | Predictor |
| est, ger, ire, ita, spa, swi, uk | Countries involved in the study (Estonia, Germany, Ireland, Italy, Spain, Switzerland, UK) | Countries | MCA 1, MCA 2 | Predictor |
| use_inc_no | No use with incentives | Use | MCA 1 | Response |
| use_no_inc_no | No use without incentives | Use | MCA 1 | Response |
| use_inc_yes_c | Use with incentives and with extra costs | Use | MCA 1 | Response |
| use_inc_yes_nc | Use with incentives and without extra costs | Use | MCA 1 | Response |
| use_no_inc_yes_c | Use without incentives and with extra costs | Use | MCA 1 | Response |
| use_no_inc_yes_nc | Use without incentives and without extra costs | Use | MCA 1 | Response |
| freq_inc_reg_irr | Regular to irregular use with incentives | Use frequency | MCA 2 | Response |
| freq_no_inc_reg_irr | Regular to irregular use without incentives | Use frequency | MCA 2 | Response |
| freq_inc_lim_no | limited to no use with incentives | Use frequency | MCA 2 | Response |
| freq_no_inc_lim_no | limited to no use without incentives | Use frequency | MCA 2 | Response |

**Table 5. Final usable response rate[1] by country.**

| Country | Respondents (n) | Respondents (%) |
|---|---:|---:|
| Estonia | 32 | 6.8 |
| Germany | 33 | 7.0 |
| Ireland | 115 | 24.3 |
| Italy | 66 | 13.9 |
| Spain | 40 | 8.4 |
| Switzerland | 52 | 11.0 |
| UK | 136 | 28.7 |
| **Total responses** | **474** | |

[1]Individual survey completion rate $\geq$ 97%.

## Knowledge exchange

There were notable differences in the rate at which beekeepers stated they communicate with growers. Overall, more than 40% asserted they never do it, particularly in the UK (68%), Ireland (63%) and Germany (39%). By contrast, 27% stated they communicate with growers more than twice a year– 58% in Switzerland, 47% in Spain, and 41% in Italy, with only 17% and 12% in Ireland and the UK respectively. Finally, about 40% Estonian beekeepers claimed to engage in communication with growers once or twice a year, with more than 20% reporting a more frequent communication.

Across all seven countries, beekeeping associations were consistently viewed as the most important sources of information on bee health, with nearly 80% of respondents reporting them as extremely or very important sources; only 1.9% thought they were not important. Beekeeping associations as important sources were followed by other beekeepers (74%), and training in person (73%). The former was particularly relevant in Ireland and Switzerland, while the latter was more important in Italy, Switzerland, and Spain. In contrast, NGOs and TV/Radio were considered not at all important by more than 30% respondents across countries.

## Perceptions of the Bee Health Card

When asked about prospective benefits of the BHC, there was a high degree of neutral opinions among respondents. The most and least perceived benefits were, respectively, increased bee health (69%) and enhanced crop pollination (32%, still almost a third of respondents) (Fig 2). Moreover, about 20% respondents disagreed with the suggestion that the BHC could reduce treatment costs for colonies, although very few expressed disagreements with any benefit at all (Fig 2). These trends were broadly consistent across countries, though beekeepers in Spain and Italy were more likely to express agreement than in other countries (Table K in S3 Appendix).

Most respondents (65%) agreed with the statement that costs could be a barrier to using the BHC–in line with percentages of disagreement for lower treatment costs as a benefit–followed by poor communication with growers (61%) (Fig 3). At a country-level, perceived BHC cost was the strongest barrier for beekeepers in the UK, Ireland, Estonia, and Switzerland, while poor communication with growers was the most predominant in Italy and Spain (Table L in S3 Appendix). By contrast, many respondents were aware of the importance of using the tool, and not concerned that it would turn out to be difficult to use; 41% and 43% disagreed with the statements 'I am not aware of the importance of using it' and 'it seems difficult to use', respectively (Fig 3).

**Table 6. Overview of total costs associated with bee health card use in each country on a per use and total national scale.** For comparison, the budget available in 2024 for bee health under the EU National Apicultural Programmes (total amount allocated to technical assistance for beekeepers and combating beehive invaders and diseases) is also included.

| Country | Number of colonies | High adoption rate (best-case scenario) | Low adoption rate (worst-case scenario) | Costs per use € (low-high adoption) | Total cost € (low-high adoption) | 2024 budget for bee health € |
|---|---|---|---|---|---|---|
| Estonia | 48,720 | 97% | 34% | €74-€64 | €0.1M-€0.3M | €0.2M |
| Germany | 771,850 | 73% | 45% | €67-€67 | €3.5M-€5.6M | €1.7M |
| Ireland | 22,278 | 94% | 50% | €91-€76 | €0.1M-€0.2M | €0.02M |
| Italy | 423,144 | 94% | 45% | €56-€55 | €1.4M-€2.9M | €6.6M |
| Spain | 2,901,680 | 90% | 55% | €51-€49 | €0.8M-€1.3M | €11.2M |
| Switzerland | 179,473 | 81% | 46% | €64-€62 | €0.5M-€0.9M | NA |
| UK | 255,000 | 90% | 46% | €48-€47 | €0.9M-€1.7M | NA |

Despite the BHC being currently under development, nearly 43% respondents stated to be moderately confident in its effectiveness, in accordance with the fact that only 10% strongly agreed that the lack of effectiveness of the BHC could represent a barrier to its use (Fig 3). Interestingly, more than 30% respondents in Germany and Italy were extremely confident that

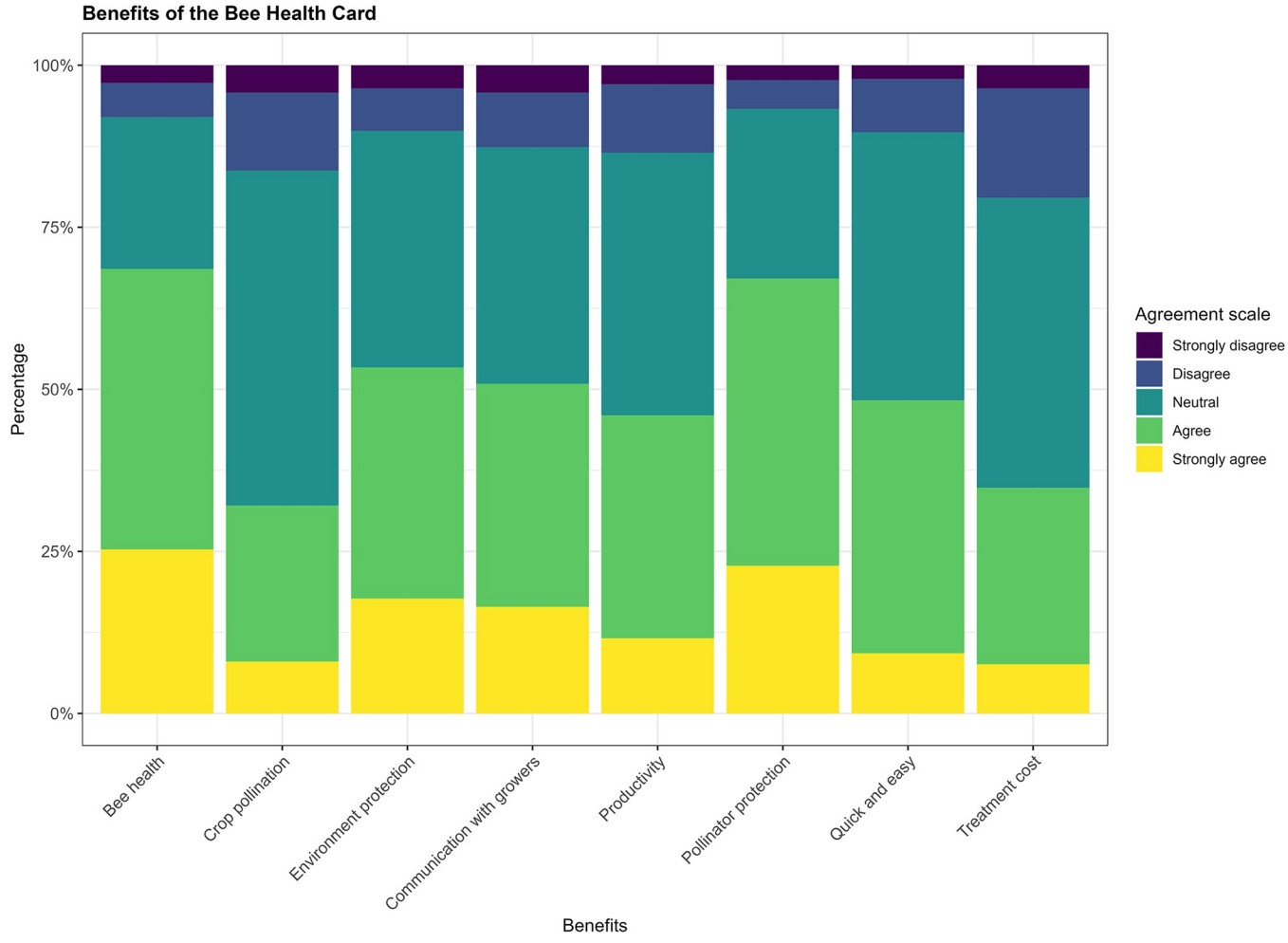

**Fig 2. Respondents' level of agreement, across countries, with prospective benefits of the Bee Health Card.**

**Barriers to the Bee Health Card**

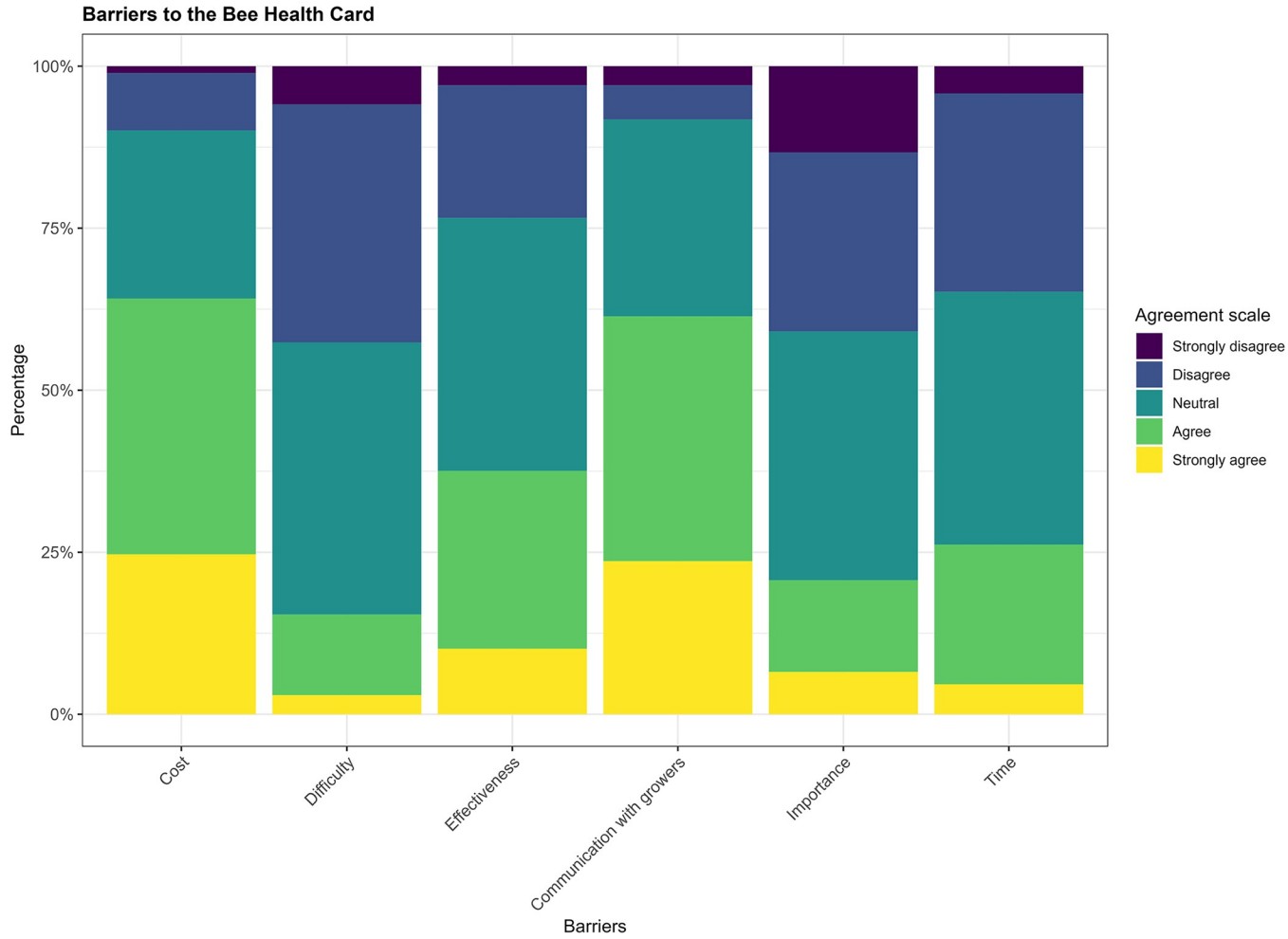

**Fig 3. Respondents' level of agreement, across countries, with prospective barriers of the Bee Health Card.**

the BHC would come to be effective, and only few stated not to be confident at all–ranging from 12% in the UK to 3% in Estonia (Table L in S3 Appendix).

## Use of the Bee Health Card

The first MCA (MCA 1) showed that, while the cluster including benefits (cluster 1) was clearly grouped the same way for agreement, disagreement, and neutral answers, this was not the case for the six barrier variables. In fact, although there was a strong clustering of time, importance, and difficulty for agree, neutral, and disagree answers, poor communication with growers, cost, and lack of effectiveness behaved differently, and did not always group with the rest of the barrier variables (Fig 4). Therefore, to account for potential differences between such barriers, only one cluster was built with time, difficulty, and importance (cluster 2), while the remaining three barriers were treated as separate variables (Table 4, Fig 4).

On MCA 1, 'evc' (*i.e.*, people extremely or very confident in tool effectiveness) was located very close to agreement with benefits and disagreement with barriers (top left quadrant), while the lack of confidence in its effectiveness ('nc') appeared to be associated with disagreement with tool benefits (top right quadrant). Instead, 'msc' (*i.e.*, people moderately to slightly confident) was situated near neutral answers related to both benefits and barriers.

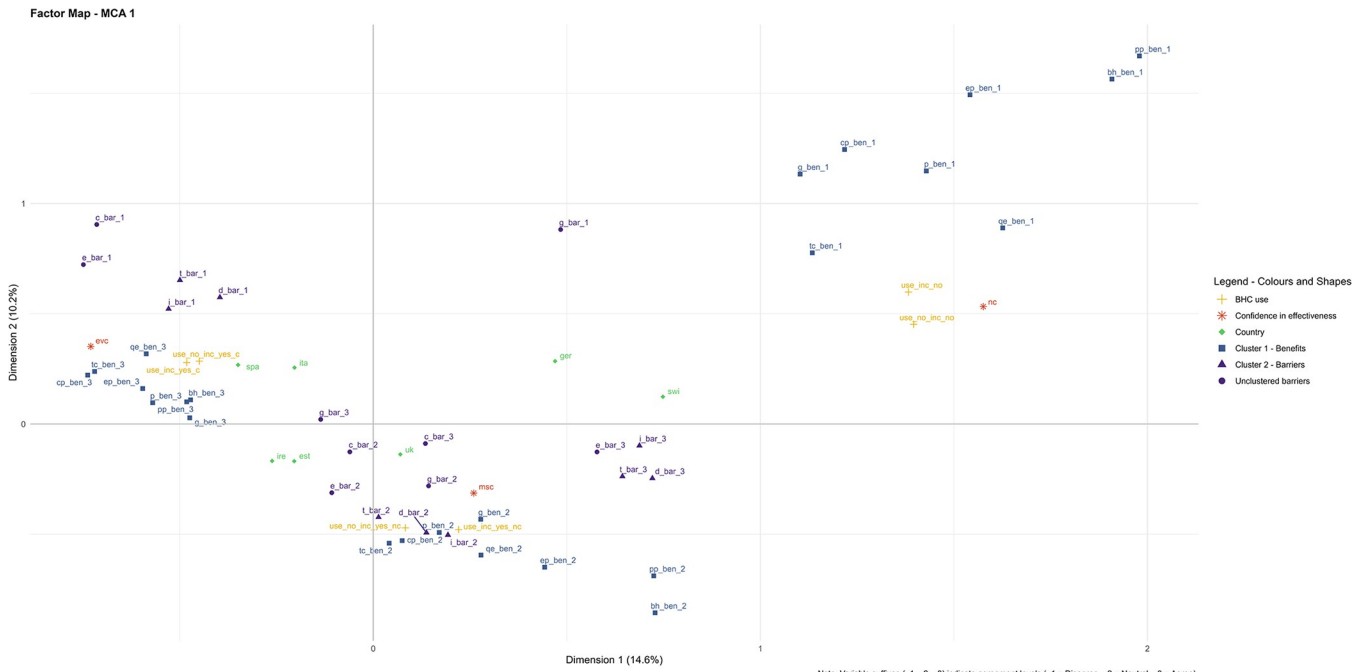

**Fig 4. MCA 1: Willingness to use the tool and accept extra costs with and without economic incentives.** A total of 58 variables were included, explaining a comprehensive 24.8% of the variation in the data (14.6% by component 1 and 10.2% by component 2). Similarities between individuals are shown through proximity on the map. Variables: **(i) Benefits:** cp_ben (crop production), bh_ben (bee health), g_ben (grower communication), qe_ben (tool quick and easy), ep_ben (environment protection), pp_ben (pollinator protection), p_ben (productivity), tc_ben (treatment cost); **(ii) Barriers:** t_bar (time-consuming), d_bar (difficult), i_bar (not important), c_bar (cost), e_bar (not effective), g_bar (poor grower communication); **(iii) Confidence in effectiveness:** evc (extremely to very confident), msc (moderately to slightly confident), nc (not confident); **(iv) Countries:** est, ger, ire, ita, spa, swi, uk; **(v) Use:** use_inc_no (no use with incentives), use_no_inc_no (no use without incentives), use_inc_yes_c (use with incentives and with extra costs), use_inc_yes_nc (use with incentives and without extra costs), use_no_inc_yes_c (use without incentives and with extra costs), use_no_inc_yes_nc (use without incentives and without extra costs). See Table 4 for in-detail description of variables and corresponding codes.

Moreover, respondents willing to use the tool with extra costs (with or without incentives) were found proximal to 'evc'. On the contrary, people willing to use the tool only without extra costs (with or without incentives) were close to 'msc', and those not willing to use the tool (with or without incentives) were nearby 'nc'. This suggests that confidence in tool effectiveness may play a pivotal role in the willingness to use the BHC and accepting its extra costs, expecting it to be statistically significant in the subsequent analyses.

In fact, the regression models showed that, with economic incentives, beekeepers were indeed more likely to use the tool when they had a higher confidence in its effectiveness and when agreeing with its benefits ($\chi^2$ = 22.05, df = 2, p<0.001 and $\chi^2$ = 7.24, df = 2, p = 0.03 respectively). It was also shown that respondents based in Germany had a lower likelihood of using the tool compared to respondents in the UK ($\chi^2$ = 12.93, df = 6, p = 0.04) (Table A in S5 Appendix).

Moreover, as expected, the probability of accepting extra costs significantly increased with the perceived level of confidence in the effectiveness of the BHC ($\chi^2$ = 14.80, df = 2, p<0.001), and with the level of agreement for its benefits ($\chi^2$ = 22.81, df = 2, p<0.001). On the contrary, it appeared to decrease for respondents agreeing with time, difficulty, and importance as barriers (cluster 2, $\chi^2$ = 19.34, df = 2, p<0.001), and with poor communication with growers ($\chi^2$ = 10.40, df = 2, p = 0.005) (Table B in S5 Appendix).

Without economic incentives, the willingness to use the tool still increased with a higher confidence in the effectiveness of the BHC and a tendency to agree with its benefits ($\chi^2$ =

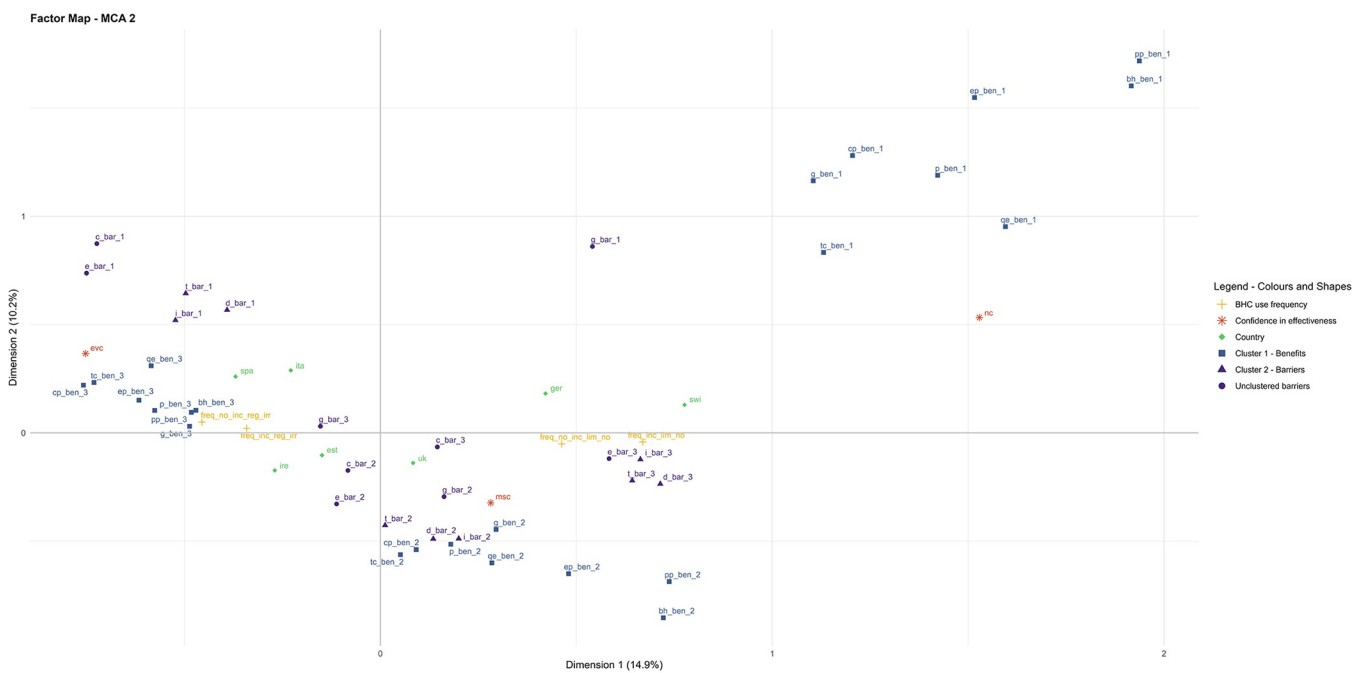

**Fig 5. MCA 2: Frequency of use of the tool with and without economic incentives.** A total of 56 variables were included, explaining a comprehensive 25.1% of the variation in the data (14.9% by component 1 and 10.2% by component 2). Similarities between individuals are shown through proximity on the map. Variables: **(i) Benefits:** cp_ben (crop production), bh_ben (bee health), g_ben (grower communication), qe_ben (tool quick and easy), ep_ben (environment protection), pp_ben (pollinator protection), p_ben (productivity), tc_ben (treatment cost); **(ii) Barriers:** t_bar (time-consuming), d_bar (difficult), i_bar (not important), c_bar (cost), e_bar (not effective), g_bar (no grower communication); **(iii) Confidence in effectiveness:** evc (extremely to very confident), msc (moderately to slightly confident), nc (not confident); **(iv) Countries:** est, ger, ire, ita, spa, swi, uk; **(v) Use frequency:** freq_inc_reg_irr (regular to irregular use with incentives), freq_no_inc_reg_irr (regular to irregular use with no incentives), freq_inc_lim_no (limited to no use with incentives), freq_no_inc_lim_no (limited to no use without incentives). See Table 4 for in-detail description of variables and corresponding codes.

26.19, df = 2, p<0.001 and $\chi^2$ = 15.37, df = 2, p<0.001 respectively) (Table C in S5 Appendix). The level of confidence in the effectiveness of the tool was also statistically significant in driving beekeepers' acceptance of extra costs ($\chi^2$ = 13.33, df = 2, p = 0.001), together with agreeing with tool benefits ($\chi^2$ = 18.32, df = 2, p<0.001). Moreover, agreement with poor communication with growers ($\chi^2$ = 9.99, df = 2, p = 0.007) and cluster 2 barriers (*i.e.*, time, difficulty, and importance, $\chi^2$ = 10.70, df = 2, p = 0.005) decreased the likelihood of accepting extra costs. Contrary to the scenario with economic incentives, beekeepers who regarded cost as a potential barrier were also less likely to accept extra costs ($\chi^2$ = 7.39, df = 2, p = 0.03) (Table D in S5 Appendix).

## Frequency of use of the Bee Health Card

In the second MCA conducted on variables related to the frequency of use of the tool in a scenario with and without planned economic incentives (MCA 2), benefits and barriers grouped the same way as in MCA 1 (Table 4, Fig 5).

Once again, in line with MCA 1, in MCA 2 people extremely/very confident in the tool effectiveness ('evc') appeared to be near the agreement with tool benefits and disagreement with its barriers, while 'msc' was close to neutral answers, and 'nc' was on the vicinity of disagreement with benefits. Moreover, a more frequent use of the tool (with or without incentives) appeared to be in the same quadrant as people extremely/very confident in the tool effectiveness (top-left), while the limited or lack of use of the tool (with or without incentives)

is located in the bottom-right quadrant near the top-right one, between people with moderate/slight confidence and those with no confidence. This suggests a further pivotal role of confidence in the tool effectiveness in influencing the frequency of tool use.

Here, the regression models confirmed that, with economic incentives, a higher confidence in the tool effectiveness corresponded to a higher probability of using it more frequently each year ($\chi^2$ = 24.46, df = 2, p<0.001). Respondents agreeing with tool benefits were also linked to a higher probability of using the BHC more frequently ($\chi^2$ = 16.90, df = 2, p<0.001) (Table E in S5 Appendix).

Similarly, without economic incentives, the likelihood of more frequent tool use was linked to higher confidence levels ($\chi^2$ = 43.47, df = 2, p<0.001). Additionally, agreeing with cost and cluster 2 barriers (*i.e.*, time, difficulty, and importance) corresponded to a lower use frequency ($\chi^2$ = 8.35, df = 2, p = 0.01 and $\chi^2$ = 6.44, df = 2, p = 0.04 respectively) (Table F in S5 Appendix), supporting MCA 2 findings.

## Costs associated with the Bee Health Card

The total cost of using and managing the BHC ranged from €33.83/use in Spain to €45.66/use in Germany, with much of the variation associated with the cost of laboratory analysis and staff (Table A in S6 Appendix). In some countries, postage was also a significant expense, even assuming half the standard rate. Of these, reusable materials amounted to €4.33/beekeeper. In total, if beekeepers were expected to pay only for their sampling materials and postage, the cost of the BHC would approximately be €11.05–22.59, in line with the estimated cost in the survey infographic ($\leq$ €25).

Extrapolating these costs to the national scales, based on the relative willingness to use the tool (Table 6), the total annual cost to adopt the BHC by all beekeepers ranged from €4.2M (Germany, high adoption rate) to €92,000 (Estonia, low adoption rate) (Table 6, see Tables B, C in S6 Appendix for a breakdown). In the best-case scenario, the cost would be entirely covered by national authorities, alongside any economic incentives. In the worst-case scenario, this cost would have to be paid by the users at about €47–90 per use.

## Impacts on bee health

Using a hypothetical 50% increase in the survival of colonies deriving from use of the BHC, the number of additional colonies surviving the winter at a national scale was assessed using the linear, pessimistic, and optimistic 'efficiency frontiers' for both high (optimistic) and low (pessimistic) adoption rates reported in Table 6.

The projections highlighted the importance of understanding the relationship between uptake and effectiveness, with additional colony survival from the tool use ranging from 6% in the pessimistic-low adoption scenario, to near the maximum 50% in the optimistic-high adoption one (Tables D-F in S6 Appendix). Most notably, the high adoption rate in Germany and Switzerland appeared lower (73% and 81% respectively) than in other countries ($\geq$90%), with a 20% and 15% difference in survival increase, respectively (Table F in S6 Appendix).

## Impacts on honey production

Using a conservative estimate of 20% difference in production between a new colony and an established one, we estimated the value of increased honey production–thanks to colonies surviving the winter that would not have done so without the BHC–at €5.43 (Ireland) to €69.4 (Switzerland) per colony (Table 7). Increased to a national scale, this would equate to between €1,000 (Ireland, pessimistic efficiency frontier) and €2.5M (Germany, high adoption

**Table 7. Estimates of additional honey production arising from adoption of the bee health card.** These are upscaled using different adoption rates (from the survey) and different efficiency frontiers (optimistic and pessimistic) to bound values.

| Country | Colonies | Value of Honey | | Adoption rate | | Honey lost per colony replaced | Additional surviving colonies | | Additional honey (000 €) | |
|---|---|---|---|---|---|---|---|---|---|---|
| | | National (000 €) | € Per colony | High | Low | | High adoption, optimistic | Low adoption, pessimistic | High adoption, optimistic | Low adoption, pessimistic |
| Estonia | 48,720 | € 8,268 | € 169 | 97% | 34% | € 33.94 | 2020 | 239 | € 69 | € 8 |
| Germany | 771,850 | € 232,870 | € 302 | 73% | 45% | € 60.34 | 41431 | 9227 | € 2,500 | € 557 |
| Ireland | 22,278 | € 605 | € 27 | 94% | 50% | € 5.43 | 433 | 107 | € 2 | € 1 |
| Italy | 423,144 | € 25,095 | € 59 | 94% | 45% | € 11.86 | 18551 | 3838 | € 220 | € 46 |
| Spain | 2,901,680 | € 126,039 | € 43 | 90% | 55% | € 8.69 | 252794 | 77243 | € 2,196 | € 671 |
| Switzerland | 179,473 | € 62,280 | € 347 | 81% | 46% | € 69.40 | 6396 | 1417 | € 444 | € 98 |
| UK | 255,000 | € 72,517 | € 284 | 90% | 46% | € 56.88 | 6823 | 1476 | € 388 | € 84 |

optimistic efficiency frontier), giving a cost-benefit ratio of €0.06 (Ireland) to €1.21 (UK) per €1 spent.

## Discussion

Using an online questionnaire survey, we explored beekeepers' willingness to adopt a novel bee health technology, the molecular diagnostic Bee Health Card (BHC) tool, to provide accessible and rapid evaluation of beehive health. Our aim was to identify potential barriers and incentives to encourage its adoption by beekeepers, addressing important gaps in the literature on beekeepers' opinions and attitudes toward new technologies, and to develop a cost-benefit analysis of the BHC, exploring the tool potential in lowering financial pressures on beekeepers and enhancing bee colony health.

### Barriers to and benefits of the Bee Health Card

**Beekeepers' perspective.** Research into beekeepers' interests and attitudes is limited [22, 29–32], and their knowledge and experience of bee health is sometimes underestimated [13]. Very few studies have addressed the need to directly investigate the impact of beekeepers' knowledge on bee management practices [64], and none have investigated beekeepers' perceptions of adopting new technological tools to help improve the health of beehives. However, numerous papers (*e.g.*, see [35, 65, 66]) have explored farmers' interests in adopting new technologies, and several parallels between farmer and beekeeper attitudes emerged from our study.

As with farmers, our results indicate that beekeepers are less willing to adopt new technologies if they are perceived as difficult to use, while they are more eager to adopt them if regarded as easy to use, not time-consuming [35, 65, 67], and functionally effective [65, 68]. This is promising, considering that the MALDI BeeTyping® analysis is expected to take a very short time to process samples [36], and that BIOP and CNRS scientists, currently working on the BHC, have stated that results will be formatted in an easy-to-read way for beekeepers (based on colour codes).

Consequently, increasing the perception of effectiveness via demonstration and access to more practical information on its functions once the tool is completed, instead of relying on a hypothetical concept, will be crucial to incentivising widespread uptake [69]. Since, overall, our respondents appeared to consider beekeeping associations as the most important sources of information on the health of their bees, we can conclude that establishing a robust and direct collaboration with both national and local associations is key to help incentivise the use

of the BHC through knowledge exchange and dissemination processes. Beekeepers would also be able to build a deeper understanding of bee health and, consequently, be better able to address health concerns in their colonies. This concept is consistent with studies on farmers' perceptions toward using new technologies, reporting that farmers with higher levels of formal education are generally more dedicated to acquiring knowledge from external sources and more likely to adopt new technologies [35, 65, 69].

One of the most important barriers to the use of the BHC, according to more than half respondents, was its potential cost. This finding is in line with the literature on farmers' perceptions of technology tools, where high costs and uncertainties regarding economic returns are key reasons for users' hesitation to adopt new technologies [35, 66, 69]. In our study, costs became a significant barrier only in scenarios with no economic incentives, where it was associated with a lower frequency of use and lower probability of accepting extra costs. Such outcome was expected, since the cost of keeping healthy hives has increased to the point of becoming unprofitable for some owners of small-scale apiaries [22, 70]. Thus, a tool that does not require further investments in time and monetary resources could alleviate some of these economic pressures, particularly on professional beekeepers who often have large-scale commercial operations [19, 22, 71].

**Incentivising Bee Health Card uptake.** Despite the widespread availability of technological tools to help enhance sustainable farming and efficiency, their adoption among farmers remains low [65]. Past studies highlighted the need of increasing knowledge in order to raise the chances of implementing new farming technologies [65, 72, 73] or to improve bee conservation [74]. Targeting farmers' knowledge of bees and crop pollination may help increase their communication with beekeepers, and also encourage them to hire healthy hives [30]. In this regard, the BHC may play a role in tackling health issues affecting the colonies, ensuring that crop yield will not be affected by reduced or impaired pollination. Although investigating farmers' willingness to pay for this new tool would contribute to our understanding with new, useful insights, we anticipate that exclusively relying on this route to support the BHC may be impractical, since farmers' aims and attitudes toward bee health tools can highly vary. Instead, it may be more effective to estimate farmers' willingness to adopt practices that benefit beekeepers, such as providing open information on where and when planning to spray insecticides so that hives could be located away from crops and/or closed during and just after spraying, to reduce risks of exposure [22, 75].

Government subsidies may represent another potential funding source and have been widely proposed by European beekeepers as an incentive to expanding their operations [30]. EU support for beekeeping-related issues through national programmes between 2020–2022 amounted to €40M/year [24, 76, 77], and through other forms of government aid such as Rural Development Programmes (RDPs), whose aims include supporting innovative technologies, agricultural innovations, and national quality schemes [78, 79]. Expanding funding opportunities to subsidise the implementation of the BHC, at least initially, would help beekeepers tackle and address potential health issues with their bees.

If economic incentives, such as subsidises, could not be offered, our results indicate that increasing the understanding of wider benefits, including on pollinator and environment health [80], could make beekeepers more likely to use the BHC.

However, in its current form, the costs of the Bee Health Card would be greater than the value of additional honey produced or health management costs saved, making investment over such a large scale difficult to justify, especially where the costs of adoption at such a scale are greater than the national budgets for bee health [49–53, 81]. Postage costs are a significant driver due to the need of including cold packs to prevent biochemical and chemical alterations of samples, which significantly increase the weight of the packaging. Similarly, staff costs are

especially high due to the amount of lab time required, which could be reduced through automation. It should be emphasised, however, that other economic benefits such as the reduced costs of replacing entire hives, which may greatly exceed the costs of tool use [22], increasing pollination services to commercial crops and monitoring environmental stressors affecting bees and other pollinators, could also arise from effective use.

**Added value for biodiversity monitoring.** Although plant protection products could help increase crop yield [82–84], pesticide pressures on pollinators can potentially negatively impact on the yield and quality of crops they pollinate [85–87]. If the BHC detected a sub-optimal environment characterised by high insecticide residues, local wild bees would also be affected; reducing insecticide usage could then benefit not only managed bees, but also wild bee populations. In support of this, the literature shows examples where reducing to some extent the use of insecticides did not affect the productivity or profitability of farmlands [88–90]. Thus, monitoring exposure to individual and combinations of pesticides through the BHC could encourage the adoption of lower input management practices, benefitting both beekeeping and farming activities in lowering pressures on bees and favouring pollination.

In addition to pesticide issues, the tool could assist with accelerating the detection of parasites and pathogens in beehives, favouring containment measures. This may be particularly helpful with American and European foulbroods, which are notifiable diseases according to the World Organisation of Animal Health (https://www.woah.org/en/disease/diseases-of-bees/) and necessitate antibiotics, when permitted [91], to prevent both monetary and colony losses [92, 93]. Another example is represented by *V. destructor* infections, which require administration of chemical- or organic-based treatments to keep mite proliferation under control [94]. *V. destructor* acts as a vector and activator for viral diseases [95], which are often covert and can lead to sudden, and apparently inexplicable, collapse of bee colonies [96]. Transmission of honey bee viruses to wild bees can also occur [97–99]. Therefore, the BHC could enable timely detection of high viral levels, and consequent *V. destructor* treatment interventions, which would result in a reduction in colony losses and viral disease spill-over to wild bee populations.

According to BIOP and CNRS scientists working on the BHC, five periods of collection per year may be enough to ensure an effective and quick handling of such colony health issues spanning the bee season: (i) end of winter (~February), (ii) growth period (~April), (iii) Main season (~June), (iv) Wintering preparation (~August), and (v) Beginning of winter (~October). For an even more accurate vision, one use per month during winter (*i.e.*, approximately from October to February) is recommended. Survey results showed that, without economic incentives, beekeepers' intention to use the BHC every month decreased, while the intention to use it only with a reasonable suspicion increased (Table O in S3 Appendix). In this perspective, it is essential to incentivise the uptake of the tool leveraging on the BHC value and, if possible, economic aids for beekeepers to ensure the tool will be used at an appropriate rate.

**Limitations and further work.** This study was designed to investigate the scope of the perceived barriers and benefits to the adoption of an 'in development' omics-based bee health tool. Although the results can potentially be applied to other bee health tools, in the absence of an extensively tested, large-scale prototype, both the effectiveness and costs of the tool explored here remain hypothetical. However, lacking a comparable technology that can give insights into practical factors affecting costs (*e.g.*, economies of scale, automation of part of the workflow through robotics, etc.), there is no basis for us to challenge our cost estimates, and the true cost is likely to be lower if beekeepers are expected to pay only for the sampling kits and postage (€11.1–22.5).

Nonetheless, this study also highlights the urgent need for more detailed socio-economic research into beekeeping and its potential environmental impacts [22, 30, 80]. Alternative

economics incentives may also be more appropriate to different groups ((*e.g.*, certification schemes could be of more interest to professionals than hobby beekeepers). Such information could be integrated into more standard economic survey methods such as choice experiments [100] to examine how changing costs and/or incentive levels may directly affect respondents' willingness to use the tool. Finally, we omitted demographic questions as most existing surveys of beekeepers have suggested that the activity is dominated by older, male beekeepers [101]. Nonetheless, a more detailed investigation into why different beekeeper groups perceived different benefits from tools would give insights into how beekeepers believe the tool should be used at a larger scale. This could facilitate modelling the impacts of different levels of uptake at an EU scale, and with that, a more complete cost-benefit analysis of the tool, with and without the incentives suggested.

The study was also limited by the necessity to use social media to disseminate the survey. Despite the involvement of national and local beekeeping associations, such distribution strategy inevitably led to self-selection of participants and substantive differences in the distribution among partner countries, since a higher proportion of English-speaking beekeepers may have been reached, presumably increasing the response rate of both Ireland and the UK. Stratified sampling could increase accuracy, however, given the inconsistent requirements for beekeeper registration across Europe, this is unlikely to have been viable across all countries, and would have also been very costly, particularly as many market research agencies do not have access to the contact details of niche demographics like beekeepers. Furthermore, by recruiting beekeepers though national organisations, the sample contained (i) a high proportion of beekeepers with an interest in bee health tools and/or bee declines, and (ii) hobby beekeepers (Table B in S3 Appendix) who, although they made up the majority of beekeepers in most countries, often manage a minority of national hives, and will not have the issues with cost scaling that a professional might.

Despite these limitations, our results nonetheless represent the first assessment of beekeeper incentives and barriers towards the adoption of health technologies. Crucially, our findings indicate that beekeepers recognised the responsibility of poor beekeeping in the decline of honeybee populations [11, 102], and have an interest in technologies to support their health management strategies, and reduce the risk of disease transmission [103]. Therefore, future research could explore (*e.g.*, through Social Network Analysis [104]) the effects that exposure to beekeeping associations, training courses, and social media will have on the adoption of novel health technology and knowledge exchange around bee health.

## Conclusions and future directions

The survey key findings underpin four broad conclusions and related recommendations.

In terms of incentivising the use of the tool, we identified the importance of enhancing beekeepers' confidence level in its effectiveness. The fact that respondents seemed to recognise the potential for the BHC to improve bee health is promising, given that the tool is still under development and that no opportunity to test it was available for them. In this regard, practical demonstrations should be performed once the tool is ready to make sure it is quick and easy to use and that no additional knowledge is required to implement it.

To establish frequent use of the tool, particularly if extra costs are to be involved, we also recommend developing well planned and targeted economic incentives to decrease economic pressures on beekeepers. Additionally, wider environmental benefits may also hold good potential; safeguarding pollinators and the environment could in fact lead to less bee health concerns, and consequently less interventions to tackle them.

To maximise knowledge dissemination, we suggest relying on beekeeping associations; our survey highlighted the importance given by beekeepers to often small, independently-run organisations in providing information on beehive health, and such organisations should be the primary focus of efforts to maximise knowledge exchange. Training workshops to directly increase beekeeper knowledge and skills should also be encouraged.

If widely adopted, the BHC could potentially be relatively inexpensive and economically beneficial to both beekeepers and farmers for pollinator-dependent crops. Although the estimated costs per use may vary among countries (€33–46), beekeepers could end up paying approximately €11–23 if the tool was even only partially subsidised. If effective, the BHC could add value to beekeeping through avoiding colony replacement expenses, thereby relieving financial pressures on beekeepers. Moreover, in addition to honey bee colonies, the BHC could also benefit other pollinators in the area, including other managed bee species and wild bee populations, as a measure of quality control of the environment in terms of pesticides, parasites, and other biotic and abiotic stressors.

## Supporting information

**S1 Appendix. Questionnaire survey questions and distribution.**
(DOCX)

**S2 Appendix. Materials required for the Bee Health Card.**
(DOCX)

**S3 Appendix. Survey results.**
(DOCX)

**S4 Appendix. Model selection.**
(DOCX)

**S5 Appendix. Regression models.**
(DOCX)

**S6 Appendix. Cost of using and managing the Bee Health Card.**
(DOCX)

## Acknowledgments

The authors wish to thank all the expert scientists and beekeepers who peer reviewed the survey, and the national beekeeping organisations of Estonia, Germany, Ireland, Italy, Spain, Switzerland, and the UK who disseminated it.

## Author Contributions

**Conceptualization:** Elena Cini, Simon G. Potts, Deepa Senapathi, Tom D. Breeze.

**Data curation:** Elena Cini.

**Formal analysis:** Elena Cini, Tom D. Breeze.

**Funding acquisition:** Simon G. Potts.

**Investigation:** Elena Cini, Matthias Albrecht, Karim Arafah, Dalel Askri, Michel Bocquet, Philippe Bulet, Cecilia Costa, Pilar De la Rúa, Alexandra-Maria Klein, Anina Knauer, Marika Mänd, Risto Raimets, Oliver Schweiger, Jane C. Stout, Tom D. Breeze.

**Methodology:** Elena Cini, Tom D. Breeze.

**Project administration:** Elena Cini.

**Resources:** Elena Cini, Matthias Albrecht, Karim Arafah, Dalel Askri, Michel Bocquet, Philippe Bulet, Cecilia Costa, Pilar De la Rúa, Alexandra-Maria Klein, Anina Knauer, Marika Mänd, Risto Raimets, Oliver Schweiger, Jane C. Stout, Tom D. Breeze.

**Software:** Elena Cini, Tom D. Breeze.

**Supervision:** Elena Cini, Simon G. Potts, Deepa Senapathi, Tom D. Breeze.

**Validation:** Elena Cini, Simon G. Potts, Deepa Senapathi, Tom D. Breeze.

**Visualization:** Elena Cini, Tom D. Breeze.

**Writing – original draft:** Elena Cini, Tom D. Breeze.

**Writing – review & editing:** Elena Cini, Simon G. Potts, Deepa Senapathi, Matthias Albrecht, Karim Arafah, Dalel Askri, Michel Bocquet, Philippe Bulet, Cecilia Costa, Pilar De la Rúa, Alexandra-Maria Klein, Anina Knauer, Marika Mänd, Risto Raimets, Oliver Schweiger, Jane C. Stout, Tom D. Breeze.

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
