## [Decision Letter · Decision Letter 0]

24 Jun 2024

PONE-D-23-41581Beekeepers’ perceptions toward a new omics tool for monitoring bee health in EuropePLOS ONE

Dear Dr. Breeze,

Thank you for submitting your manuscript to PLOS ONE. After careful consideration, we feel that it has merit but does not fully meet PLOS ONE’s publication criteria as it currently stands. Therefore, we invite you to submit a revised version of the manuscript that addresses the points raised during the review process.

First, I would like to apologize for the somewhat lengthy review process. Since the first two reviewers were quite contrary in their recommendations, I decided to involve a third reviewer. As it turned out, this was very much to the benefit of the authors. You can find the detailed reviews below. In my view, the most important points are as follows: If possible, more information about the respondents should be provided (age distribution, gender, education level, etc.). It should be made very clear exactly what material was available to the respondents. Furthermore, information on what maximum costs would be acceptable for a beekeeper would be helpful.

We look forward to receiving your revised manuscript.

Kind regards,

Wolfgang Blenau

Academic Editor

PLOS ONE

2. We note that Figure 1 in your submission contain copyrighted images. All PLOS content is published under the Creative Commons Attribution License (CC BY 4.0), which means that the manuscript, images, and Supporting Information files will be freely available online, and any third party is permitted to access, download, copy, distribute, and use these materials in any way, even commercially, with proper attribution. For more information, see our copyright guidelines: http://journals.plos.org/plosone/s/licenses-and-copyright.

Reviewers' comments:

Reviewer's Responses to Questions

**Comments to the Author**

1. Is the manuscript technically sound, and do the data support the conclusions?

Reviewer #1: Yes

Reviewer #2: Partly

Reviewer #3: Yes

2. Has the statistical analysis been performed appropriately and rigorously? 

Reviewer #1: Yes

Reviewer #2: Yes

Reviewer #3: Yes

3. Have the authors made all data underlying the findings in their manuscript fully available?

Reviewer #1: Yes

Reviewer #2: Yes

Reviewer #3: Yes

4. Is the manuscript presented in an intelligible fashion and written in standard English?

Reviewer #1: Yes

Reviewer #2: Yes

Reviewer #3: Yes

5. Review Comments to the Author

Reviewer #1: The manuscript “Beekeepers’ perceptions toward a new omics tool for monitoring bee health in Europe” by Cini et al. submitted to PLoS One (PONE-D-23-41581) deals with the analysis of a beekeeper survey related to the willingness of adoption of the Bee health Card (BHC). An online survey was conducted with participants originating from seven European countries. The sample size is okay, the analysis as well, with some interesting results that can help in the implementation of the BHC in the future. However, I have a few remarks, that should be taken into consideration during a revision of the manuscript.

What is known about the respondents? How is the age distribution, gender, etc? Is this more or less equal among countries? It is known that adoption of new technologies is also gender-specific as females tend to adopt new technologies easily when at low cost while males adopt are the adopters of cost-intensive technologies when they are convinced of the technology, it seems to be crucial to have information about gender. Similar to this is age as younger people might be more technology affine while older people are more reserved about new technologies. In a similar way, the level of formal education might play a role, but also the time of experience with beekeeping. Thus, I think this is crucial information that should be 1. Presented as an overview per country (e.g. in table 5), and 2. incorporated into the analysis to be able to see how interventions and incentives could be tailor-made to address specific target groups.

L48/49 also their recruitment behaviour contributes to effective pollination, especially in mass-flowering crops

L53-58 I guess the loss of suitable habitat for nesting sites of solitary bees (e.g. most are soil nesting) has also some impact on pollination.

L100-101 might be worthwhile to mention that amongst the listed countries the largest honey producing countries as well as the largest agricultural producers are included

L119 introduce abbreviation ECB in brackets as you use it later on (e.g. L121)

Fig 1. Icon of beekeeper and analytical laboratory is wrong and needs to be changed.

L395-398 did you ask for the level of formal education of the beekeepers? Is it the same as with farmers?

L439-441 why cold packs need to be included? Isn’t it possible to transport samples in a buffer solution that resembles RNAlater (like a “self-made” high salt solution, that is distributed to beekepers)?

L460-465 Varroa control during regular hive inspection is so easy due to the use of the sugar or alcohol shake method, that the bee health card will not help here. Besides that, the use of acaricides is not the only solution, many beekeepers also use organic acids as treatment or prophylaxis.

L495ff why the study was limited from the beginning on to social media? Why not going through the national and/or local beekeeping associations? What is the advantage of going social media only? I guess it is a “quick-and-dirty” approach, it is fast and cheap and you might get a few respondents, but their demographics etc is biased.

Ref 39 incomplete (please, check for all references, I just came across this one but didn't check all).

Reviewer #2: Cini et al evaluated beekeepers’ perception of the use of the “Bee Health Card” program for evaluating honey bee colony health, and the potential economic benefits of the use of this program in seven countries. Understanding the interests, needs, incentives and barriers to adoption of such tools by stakeholder groups is essential, but often this research is not being conducted. Thus, such a study is very valuable both in its outcomes and its approach, which can be potentially be used by other researchers to evaluate other tools and stakeholders.

Unfortunately, the manuscript does not provide enough context for this tool and how it could be used, and thus the analyses and results are confusing. Below are some suggestions for addressing these issues. Overall, however, it seems that the study results are specifically informative for the developers of the Bee Health Card tool, and there is little information that can be used to inform the development of other tools or the development of assessments of other tools. For example, it seems that a main barrier for the beekeepers to use this tool is cost, but there is no information about what maximum cost would be acceptable to a beekeeper (which would help others in future tool design projects). Additionally, beekeepers whose motivation to participate in beekeeping (survey question 3) may differ in their likelihood to pay for the Bee Health Card or other such services, but this information is not included. Similarly, were beekeepers who were more likely to conducted detailed checks on their colonies (Q5) more likely to be interested in using the Bee Health Card? Including these kinds of analyses would help this study be more broadly relevant and informative.

Additional points:

Was the only information given to the beekeepers the infographic shown in Figure 1? Were the beekeepers provided any information about how the information they obtained could be used to design treatments or management strategies?

If the beekeepers were provided only with the infographic, it is difficult to see how they could develop any strong opinion of it. Is there a way to determine if the responses did not simply represent “chance”? For example, in Figure 2 and 3, the beekeepers were asked what the benefits or barriers to adopting the tool were, and given several benefits/barriers with which they could “strongly agree, agree” etc. The answers to all of these questions is largely the same. As described, it is difficult to know how the tool could support “better communication with growers” or “environmental protection”, but ~50-60% of beekeepers strongly agreed/agreed with this.

In Table 5, what is the % of respondents mean? What this the % of respondents whose surveys were usable? If so, please include the number of people who actually completed the survey here.

Line 245: Most of the respondents seem to be hobbyist beekeepers and thus would not be working with growers on pollination contracts. It would be helpful to report how the different groups (hobbyists and professionals) described their level of communication with growers.

Line 268-272: 61% of beekeeper stated felt that “lack of communication with growers” was a barrier to using this tool (Q13). Again, this is confusing – why do beekeepers need to communicate with the growers to use the tool? Is the idea that the analysis would identify pesticides in the bee samples and then the beekeepers would ask the growers to stop using these pesticides? So, the lack of communication with the growers is a barriers to effectively implementing the data derived from the tool?

Overall, the MCA analysis is confusing to interpret. It seems that the main outcome is the likelihood of using the tool is positively correlated with belief that the tool will be effective and/or the economic benefit of using the tool. Is there another way the authors could demonstrate this? It seems that some of the specific answers to the survey questions - perhaps separated according to the beekeepers’ interests in beekeeping or their current practices - would be more informative than this MCA analysis in terms of interpreting the data and information future tool development.

Q17 of the survey asked how often beekeepers would be willing to pay for the assessment. How did this compare with the manufacturer’s recommendation that the assessment be conducted 5 or more times a year?

Reviewer #3: In the paper "Beekeepers’ perceptions toward a new omics tool for monitoring bee health in Europe" the Authors investigate beekeepers’ willingness to adopt the Bee Health Card, a molecular diagnostic tool able to rapidly assess bee health.

The design and the logic of the paper seem well established and worth publication. The sampling process is adequately done and does provide useful information to beekeeping.

I only have a couple of minor revisions to be addressed:

- since Italy has been one of the countries selected for this study, I suggest to cite Vercelli et al. 2021_Insects, who addressed beekeepers’ perceptions and the main issues affecting the sector through focus group discussions.

- in the Introduction only the Varroa is reported to be a main phytosanitary issue for bees. I would add the major problem of the accidental introduction of invasive exotic species as the yellow-legged hornet, Vespa velutina, recently introduced into Europe from Asia.

6. PLOS authors have the option to publish the peer review history of their article (what does this mean?). If published, this will include your full peer review and any attached files.

Reviewer #1: No

Reviewer #2: No

Reviewer #3: No

---

## [Decision Letter · Decision Letter 1]

24 Sep 2024

PONE-D-23-41581R1Beekeepers’ perceptions toward a new omics tool for monitoring bee health in Europe

PLOS ONE

Dear Dr. Breeze,

Thank you for submitting your manuscript to PLOS ONE. After careful consideration, we feel that it has merit but does not fully meet PLOS ONE’s publication criteria as it currently stands. Therefore, we invite you to submit a revised version of the manuscript that addresses the points raised during the review process.

The Academic Editor who originally assigned themselves to this manuscript became unavailable, and I am now the Academic Editor. I do apologize on behalf of the journal office for the delay this has caused. While Reviewers #1 and #3 provided generally positive feedback and found the manuscript suitable for publication, I share some concerns raised by Reviewer #2: insufficient background on the tool, excessive focus on BHC marketing, limited analysis of other interesting results, and a somewhat confusing presentation of the MCA results.

In particular, the manuscript can be improved by addressing the following** required revisions**:

**Adding more details in the introduction section about the tool.** While the tool is briefly described, it is not immediately clear whether it is a field or laboratory test. Please specify that it is a laboratory test to avoid confusion, particularly for readers unfamiliar with MALDI. Additionally, the potential for BHC detection should be detailed—what pathogens and parasites does it target? What types of pesticides? I inderstand that the techonology is under development but some hints on the range of detection may help. Does the method involve sending live bees? If so, why are cold packs recommended (L430-432)? Please clarify this in the text.**Providing a clearer description of the MCA results. **The explanation of how clustering is performed is confusing. For example, in lines 283-284: “Therefore, to account for potential differences between such benefits, two different clusters were built based on the way the disagrees and neutrals grouped on the map (cluster 1a and cluster 1b) (Table 4, Fig 4).” However, the cited clusters represent benefits to which beekeepers disagree only, not neutral ones. Moreover, cluster 1a is composed of three variables that are interspersed with other variables not included in the cluster. This clustering approach and the way the MCA results are described seem arbitrary. This section should be more descriptive of what the data indicates, in a linear and clearer way (e.g. see MCA results in, https://doi.org/10.1371/journal.pone.0230999). The most important, and probably the first, thing to say here should be what is the relationship between the willingness to adopt the tools and the other variables. For example, in fig. 4 “evc” clusters together with ninc and inc, suggesting that confident beekeepers may adopt the tools regardless of incentives. Is this commented in the manuscript? Moreover, cluster formed around “evc” seems to be formed by agreement with benefits, rather by disagreement with barriers (cluster 2a), while cluster formed around “nc” seems formed equally by agreement with barriers and disagreement with benefits. Again, this seems not discussed in the manuscript. Similarly, disagreement in increased production groups together with other benefits agreement related to confidence in the tool, suggesting that confident beekeepers would use the tool regardless of its primary effect on production. These kinds of considerations should emerge from results and properly discussed in a revised version of the manuscript.**Improving the clarity of MCA figures. **The MCA figures use acronyms that are defined in Table 4, which is located in a different section of the manuscript, making the figures difficult to interpret (I had to open three windows simultaneously to understand the results). To enhance the figures, I suggest either a) including the acronym explanations directly in the figure or its legend, or b) using more descriptive labels for the variables.

**Moreover, I recommend to try adding a broader perspective on beekeepers’ perception of new technologies. **The manuscript would benefit from presenting other collected data which may be related to willingness to use BHC, such as responses on bee decline and beekeeper information exchange. Another intriguing, but seemingly undiscussed, aspect is the open question on which main health issues BHC should be able to detect (Table P). Are “confident” beekeepers more interested in detecting particular health issues? By properly presenting and discussing these findings, the authors would significantly enhance the manuscript’s contribution to understanding beekeepers’ perceptions of new tools.

We look forward to receiving your revised manuscript.

Kind regards,

Andrea Becchimanzi

Academic Editor

PLOS ONE

Reviewers' comments:

Reviewer's Responses to Questions

**Comments to the Author**

1. If the authors have adequately addressed your comments raised in a previous round of review and you feel that this manuscript is now acceptable for publication, you may indicate that here to bypass the “Comments to the Author” section, enter your conflict of interest statement in the “Confidential to Editor” section, and submit your "Accept" recommendation.

Reviewer #1: All comments have been addressed

Reviewer #2: All comments have been addressed

2. Is the manuscript technically sound, and do the data support the conclusions?

Reviewer #1: Yes

Reviewer #2: No

3. Has the statistical analysis been performed appropriately and rigorously? 

Reviewer #1: Yes

Reviewer #2: Yes

4. Have the authors made all data underlying the findings in their manuscript fully available?

Reviewer #1: Yes

Reviewer #2: Yes

5. Is the manuscript presented in an intelligible fashion and written in standard English?

Reviewer #1: Yes

Reviewer #2: Yes

6. Review Comments to the Author

Reviewer #1: (No Response)

Reviewer #2: I appreciate the authors taking the time to respond to my previous review. Unfortunately, based on their responses, it seems that this survey was conducted at a point when there was too little information about the tool to be able to properly represent it to beekeepers, to design appropriate questions for beekeepers, or to recruit the appropriate set of beekeepers to complete the survey. Indeed the authors noted "Estimating the maximum willingness to pay for the tool was considered, however we eventually did not pursue this route for a number of reasons: 1) The tool description was not detailed enough for a reliable product description, which would have casted considerable doubt on the validity of the results..."

As previously noted, there are several illogical responses in the survey – for example, beekeepers feeling the tool use is limited by “lack of communication with the growers” when there is nothing about the tool that requires communication with growers - I appreciate the authors' comments that it would be helpful for growers to tell beekeepers when they are spraying pesticides so they can close their hive entrances, but this does not relate to whether a beekeeper can use the tool. Additionally, the rationale for not asking for basic demographic information from the beekeepers or evaluating the cost the beekeepers were willing to pay for the tool is not sufficient (would it really be that difficult to analyze the data from these questions?). It is clear from the text that the cost of the tool (materials, shipping and processing) will be much higher than the 25€ quoted in the survey, and having this information would be valuable for the group designing the tool (or any entity that would design such diagnostic services) to know if it will even be marketable, particularly if the recommendation is for the tool to be used repeatedly throughout the year. My concern with publishing the current study in PLOS ONE is that it will lead to other poorly designed tools and surveys in the future, since others may model their research on this manuscript.

7. PLOS authors have the option to publish the peer review history of their article (what does this mean?). If published, this will include your full peer review and any attached files.

Reviewer #1: No

Reviewer #2: No

---

## [Author Response · Author response to Decision Letter 1]

10 Dec 2024

See the revised response to reviewers

---

## [Editor Report · Decision Letter 2]

15 Dec 2024

Beekeepers’ perceptions toward a new omics tool for monitoring bee health in Europe

PONE-D-23-41581R2

Dear Dr. Breeze,

We’re pleased to inform you that your manuscript has been judged scientifically suitable for publication and will be formally accepted for publication once it meets all outstanding technical requirements.

Kind regards,

Andrea Becchimanzi

Academic Editor

PLOS ONE
---

## [Editor Report · Acceptance letter]

23 Dec 2024

PONE-D-23-41581R2 

PLOS ONE

Dear Dr. Breeze, 

I'm pleased to inform you that your manuscript has been deemed suitable for publication in PLOS ONE. Congratulations! Your manuscript is now being handed over to our production team.

Kind regards, 

on behalf of

Dr. Andrea Becchimanzi 

Academic Editor

PLOS ONE